# SIOP PNET5 MB Trial: History and Concept of a Molecularly Stratified Clinical Trial of Risk-Adapted Therapies for Standard-Risk Medulloblastoma

**DOI:** 10.3390/cancers13236077

**Published:** 2021-12-02

**Authors:** Martin Mynarek, Till Milde, Laetitia Padovani, Geert O. Janssens, Robert Kwiecien, Veronique Mosseri, Steven C. Clifford, François Doz, Stefan Rutkowski

**Affiliations:** 1Department of Pediatric Hematology and Oncology, University Medical Center Hamburg-Eppendorf, 20246 Hamburg, Germany; m.mynarek@uke.de; 2Mildred Scheel Cancer Career Center HaTriCS4, University Medical Center Hamburg-Eppendorf, 20246 Hamburg, Germany; 3Hopp Children’s Cancer Center (KiTZ), 69120 Heidelberg, Germany; t.milde@kitz-heidelberg.de; 4Clinical Cooperation Unit Pediatric Oncology, German Cancer Research Center (DKFZ) and German Consortium for Translational Cancer Research (DKTK), 69120 Heidelberg, Germany; 5Department of Pediatric Oncology, Hematology and Immunology, Heidelberg University Hospital, 69120 Heidelberg, Germany; 6Oncology Radiotherapy Department, CRCM Inserm, Aix-Marseille University, UMR1068, CNRS UMR7258, AMU UM105, Genome Instability and Carcinogenesis, Assistance Publique des Hôpitaux de Marseille, 13284 Marseille, France; Laetitia.Padovani@ap-hm.fr; 7Department of Radiation Oncology, University Medical Center Utrecht, 3508 GA Utrecht, The Netherlands; g.o.r.janssens@umcutrecht.nl; 8Princess Máxima Center for Pediatric Oncology, 3584 CS Utrecht, The Netherlands; 9Institute of Biostatistics and Clinical Research, Faculty of Medicine, University of Münster, 48149 Münster, Germany; robert.kwiecien@ukmuenster.de; 10Department of Biostatistics, Institut Curie, 75248 Paris, France; veronique.mosseri@curie.fr; 11Newcastle University Centre for Cancer, Translational and Clinical Research Institute, Faculty of Medical Sciences, Newcastle University, Newcastle upon Tyne NE1 7RU, UK; s.c.clifford@ncl.ac.uk; 12SIREDO Center (Care, Innovation and Research for Children, Adolescents and Young Adults with Cancer), Institut Curie, Paris and Université de Paris, 75248 Paris, France; francois.doz@curie.fr

**Keywords:** medulloblastoma, children, trial, CNS, brain tumour

## Abstract

**Simple Summary:**

The European trial SIOP PNET5 MB was initiated in 2014 and will remain open to recruitment until 2022. It is the first European trial using clinical, histological, and molecular parameters to stratify treatments for childhood medulloblastoma, based on disease risk. In the standard-risk stratum, a randomized intensification of carboplatin concomitant to radiotherapy is investigated. In the favourable-risk stratum, defined by localized WNT subgroup disease, reduction of craniospinal radiotherapy intensity (from 24 to 18 Gy) and reduced maintenance chemotherapy is investigated for children <16 years old at diagnosis. Two additional exploratory strata (WNT-HR and SHH-TP53) have been implemented during the trial. The use of biological parameters for stratification has proven feasible in a prospective multicentre setting, and may improve future risk-adapted treatment. The primary endpoint is 3-year event-free survival. Late effects on hearing, endocrine- and neurologic function, alongside health-related quality of life (e.g., health status, behavioural outcomes), are secondary endpoints.

**Abstract:**

Background. SIOP PNET5 MB was initiated in 2014 as the first European trial using clinical, histological, and molecular parameters to stratify treatments for children and adolescents with standard-risk medulloblastoma. Methods. Stratification by upfront assessment of molecular parameters requires the timely submission of adequate tumour tissue. In the standard-risk phase-III cohort, defined by the absence of high-risk criteria (M0, R0), pathological (non-LCA), and molecular biomarkers (*MYCN* amplification in SHH–MB or *MYC* amplification), a randomized intensification by carboplatin concomitant with radiotherapy is investigated. In the LR stratum for localized WNT-activated medulloblastoma and age <16 years, a reduction of craniospinal radiotherapy dose to 18 Gy and a reduced maintenance chemotherapy are investigated. Two additional strata (WNT-HR, SHH-TP53) were implemented during the trial. Results. SIOP PNET5 MB is actively recruiting. The availability of adequate tumour tissue for upfront real-time biological assessments to assess inclusion criteria has proven feasible. Conclusion. SIOP PNET5 MB has demonstrated that implementation of biological parameters for stratification is feasible in a prospective multicentre setting, and may improve risk-adapted treatment. Comprehensive research studies may allow assessment of additional parameters, e.g., novel medulloblastoma subtypes, and identification and validation of biomarkers for the further refinement of risk-adapted treatment in the future.

## 1. Introduction and Status of Knowledge When SIOP PNET5 MB Was Planned

Medulloblastoma is a highly cellular malignant embryonal neoplasm [1]. It is the most common malignant brain tumour in children, accounting for 15 to 20% of all childhood primary central nervous system (CNS) neoplasms. Medulloblastoma arises in the posterior fossa, from the cerebellar vermis in the roof of the 4th ventricle, cerebellar hemispheres or the dorsal brainstem. Medulloblastomas have a marked propensity to metastasize via CSF pathways, and evidence of such metastatic spread is present in up to 35% of cases at diagnosis [1].

When the SIOP PNET5 MB trial was planned, the following histological variants of medulloblastoma were recognized in the WHO classification of CNS tumours [2]: classic medulloblastoma, desmoplastic/nodular medulloblastoma, medulloblastoma with extensive nodularity, large-cell medulloblastoma, and anaplastic medulloblastoma. The treatment of medulloblastoma involved surgical resection followed by radiotherapy and chemotherapy. While this combined-modality treatment regimen had substantially improved the cure rate, medulloblastoma remained incurable in approximately one-third of patients [3], and 20–25% of standard risk patients [4,5]. Moreover, survivors suffer from long-term toxic side effects related to therapy that often seriously affect their quality of life [6,7,8,9].

At that time, prognosis was most commonly assessed based on imaging and CSF cytology risk criteria. Risk-adapted treatment of medulloblastoma was established, using age, extent of resection, and presence of metastases for stratification. Patients were usually assigned to treatment groups based on absence of established imaging/cytology high-risk-factors (metastatic disease (M+, ~30% of patients), sub-total resection (R+, ~10%), and large-cell/anaplastic pathology (LCA, ~15%). Patients over 3 years of age were divided into: standard-risk group (SR-MB; negative for all clinical risk-factors, 40–50% of patients, ~80% survival) or high-risk group (HR-MB; positive for any clinical risk-factor, ~30% of patients, ~60% survival), and received risk-adapted therapies accordingly. However, these clinical schemes stratified risk inaccurately, based on the observed survival differences within each group.

In the prospective European predecessor trial HIT/SIOP-PNET4, children older than 3–5 years of age without metastatic disease were randomized to receive postoperative conventional standard radiotherapy (23.4 Gy to craniospinal axis plus boost to 54 Gy to the entire fossa posterior, 1.8 Gy per day) or hyperfractionated radiotherapy (36 Gy to craniospinal axis plus boost to 60 Gy to fossa posterior plus 8 Gy to tumour bed; 2 × 1 Gy per day), followed by eight cycles of maintenance chemotherapy with CDDP, CCNU, and vincristine. Survival rates were not significantly different between the two treatment arms: 5-year event-free survival (EFS) was 77 ± 4% in the STRT group and 78 ± 4% in the HFRT group; corresponding 5-year OS was 87 ± 3% and 85 ± 3%, respectively. A postoperative residual tumour of more than 1.5 cm^2^ was the strongest negative prognostic factor [10], with 5-year EFS of 64 ± 9% versus 82 ± 2% in the subgroup of patients with completely resected tumours. This is exactly in the same range as outcomes of contemporary clinical trials including patients with completely resected medulloblastoma treated with postoperative radiotherapy [4,5].

Age less than 3 years was globally associated with a two-fold higher risk of disease progression within 5 years of diagnosis in comparison with older patients [11,12]. One of the reasons for this less favourable prognosis was a different biological behaviour. It was known that different histological medulloblastoma variants had different age distributions, with nodular desmoplastic medulloblastomas being more frequent in infant medulloblastoma, and associated with a good prognosis. Conversely, prognosis of infants with classic medulloblastoma was worse compared to older children with classic medulloblastoma [13,14]. Decreased prognosis was also explained by an unwillingness to apply dose-intense radiotherapy in this young age group, as this causes severe damage to the developing brain [15]. Due to these factors, infants were not considered to be ‘standard risk’ patients, and could not be included into the trial SIOP PNET5 MB.

Patients with disseminated disease had a much poorer prognosis. The presence of metastatic disease at presentation as diagnosed by the presence of meningeal enhancement on MRI of the brain (Chang Stage M2) or spine (Chang Stage M3) clearly carried a poor prognosis [11,16]. Although it was not consistently used for stratification in the early trials, microscopic spread to the CSF had also been shown to be associated with an impaired prognosis, independently of the presence of macroscopic metastases (Chang Stage M1) [17,18,19,20]. It was widely accepted that patients should be staged by MRI and CSF analysis to exclude metastases in order to be regarded as standard risk patients. However, in a multicentre trial with a large number of participating centres, technique and quality of MRI imaging differs. Therefore, a standardized imaging technique was defined in the predecessor trial PNET 4, and central reference assessment of the MRI was recommended. In PNET4, the outcome of patients whose scans had not been centrally reviewed was found to be worse than of patients in which central review of MRI scans of brain and spine had taken place [10]. This suggested that quality assurance of imaging was a relevant tool for keeping the group of included patients clear of patients with falsely negative metastasis staging. The relevance of MRI review had also been shown by other groups [5,21].

Within PNET 4, residual tumour >1.5 cm^2^ was associated with an impaired prognosis [10]. Therefore, patients with residual tumours >1.5 cm^2^ were not considered as standard risk patients within PNET 5 MB. However, the definition of 1.5 cm^2^ (on MRI) as the limit for inclusion was arbitrary. Other groups used the definition of ‘any measurable’ tumour on MRI [21], and even within groups that used the definition of 1.5 cm^2^ there was no international consensus about the plane in which this area should be measured and calculated. As the early trials were based on CT imaging, the axial plain was commonly used for area calculation. But with the use of MRI, calculation of residual tumour area was also estimated in the maximum cross-sectional area, or as volume in three dimensions. To allow comparability to the earlier trials, it was decided to use 1.5 cm^2^ on the axial plane in SIOP PNET5 MB. For the estimation of the extent of residual disease, a patient’s pre-operative MRI imaging should be compared with the post-operative imaging. Adherence to per protocol standard MRI techniques are important for proper comparison between pre- and post-operative imaging. It was accepted that postoperative imaging is best performed within 72 h of surgery, after which post-operative changes render interpretation of residual disease difficult. For the purposes of the SIOP PNET5 MB trial, all patients should therefore have post-operative MRI imaging before and after contrast agent injection within 72 h of surgery.

More recent data suggested that the histological subtype of medulloblastoma as well as biological factors influence clinical behaviour, and could therefore be used to optimize treatment stratification [22,23,24].

## 2. Selection of the SIOP PNET5 MB Trial Population: Biomarker-Driven Disease Sub-Classification and Risk-Stratification

### Biomarker-Driven Sub-Classification and Risk-Stratification Schema

First extensive trial-based biological studies of medulloblastoma were performed on previous SIOP-PNET clinical trials cohorts; SIOP-UKCCSG-PNET3 [25,26,27] and HIT-SIOP-PNET4 [28,29]. Through these and other works, a series of medulloblastoma biomarkers were discovered and validated to show consistent cohort-wide prognostic relationships. Most notably, the WNT subgroup was identified first as a distinct molecular disease subgroup, which was consistently associated with a favourable prognosis (>90% survival) in multiple trials cohorts [4,27,28,29,30]. WNT tumours are rare in young children (peak around 9 years of age), and are associated with a WNT profile (on expression or DNA-methylation-based profiling), *CTNNB1* mutation, isolated chromosome 6 loss and nuclear localisation of the ß-catenin protein (encoded by *CTNNB1*). Evidence indicates that WNT subgroup tumours which arise in adults (>16.0 years at diagnosis) do not share the favourable outcomes of childhood WNT tumours [28,31].

In addition, consistently observed high-risk biological factors (*MYC* amplification (~5% of tumours), *MYCN* amplification (~10%)) were identified [30,32,33]. Critically, in the PNET3 clinical trial cohort, retrospective application of these biomarkers alongside WNT status and clinical and pathological risk-markers significantly showed that the complete cohort of non-infant patients could be separated into favourable-, standard- and high-risk groups. These findings both validated the utility of this combined clinical-molecular stratification scheme and provided a strong rationale for its prospective adoption as the basis for SIOP PNET5 MB [30,34].

The medulloblastoma scientific community reached further consensus in 2012 that the disease comprises four molecular disease subgroups—WNT, SHH, Group 3, and Group 4—characterised by their transcriptomic, methylomic and genomic signatures, with distinct clinico-pathological features and developmental origins [35]. In addition to the WNT subgroup (10–15% of tumours), the sonic hedgehog (SHH) subgroup (~25%) is characterised by mutational aberrations in the SHH pathway (e.g., *PTCH1, SUFU*) and is associated with desmoplastic/nodular histology (particularly in young children) and frequent chromosome 9q deletions. Group 3 medulloblastomas (~25%) typically have either classic or large cell/anaplastic pathology, with genomic aberrations including isochromosome 17q (i17q) and, most characteristically, *MYC* amplification. Group 4 tumours (~40%) are typically either classic or large cell/anaplastic pathology and also frequently harbour i17q [35,36].

Unlike the WNT subgroup, SHH, Group 3 and Group 4 tumours do not show clear subgroup-wide prognostic differences in trials-based cohorts of childhood MBs [28,34]. However, new prognostic biomarkers have emerged over the last 10 years, which have refined our understanding of such intra-subgroup heterogeneity. Most notably, SHH subgroup tumours with *TP53* mutations have been identified to have a particularly poor prognosis [37], with these relationships now validated across independent cohorts, including SIOP-PNET4 [28,38]. A significant proportion of this patient group harbour germline *TP53* mutations in the context of diagnosed or undiagnosed Li-Fraumeni syndrome [39]. Moreover, we discovered the poor prognosis associated with *MYCN* amplification is subgroup-specific. In both the SIOP-PNET4 clinical trial and other independent cohorts, *MYCN* amplification within the SHH subgroup is strongly associated with *TP53* mutation, LCA pathology and a very poor prognosis, whereas its detection in Group 4 tumours has no prognostic impact [28,38]. Together, these findings have supported amendments to the SIOP PNET5 MB protocol, through incorporation of SHH-*TP53* into the SIOP PNET5 MB diagnostic and stratification repertoire, and the inclusion of *MYCN*-amplified Group 4 tumours with no other high-risk features in the PNET-MB-SR stratum.

Importantly, since the inception of SIOP PNET5 MB, such contemporary molecular diagnostics have become standard-of-care. In addition to underpinning the trials described here, molecular factors (e.g., subgroup and *TP53* mutation), alongside histological variants, form the basis of medulloblastoma sub-classification in the 2016 and 2021 WHO classifications of central nervous system tumours [1,40].

## 3. Treatment

In the beginning of SIOP PNET5 MB in 2014, only patients with ‘clinically standard risk’ medulloblastoma (i.e., non-metastatic medulloblastoma not belonging to the large-cell or anaplastic histological subtype, without *MYC* or *MYCN* amplification) were included. These patients are then stratified according to the activation of the WNT pathway: patients with WNT-activated medulloblastoma are stratified to the low-risk [41] stratum (Figure 1) while patients with non-WNT medulloblastoma are stratified to the standard risk (SR) stratum (Figure 2). In 2017, patients with biologically low-risk (i.e., WNT-activated) medulloblastoma and high-risk features, as well as patients with biologically very-high-risk (i.e., SHH-activated, TP53 mutated) medulloblastoma became eligible (Figure 3; see below for details and rationale).

### 3.1. Surgery

The classical first element of therapy is maximal safe tumour resection to reduce the tumour volume as much as possible. Because of the importance of the residual tumour for progression-free survival (PFS) [10], patients with large post-operative residual tumour who are considered to reach a gross total or near total resection by second surgery may undergo second surgery before inclusion and might become eligible for participation in SIOP PNET5 MB, if a postoperative residual tumour of <1.5 cm^2^ (axial) can be achieved and postoperative can start in due time after first surgery.

### 3.2. Radiotherapy

As in PNET4 [10], post-operative therapy is to start with radiotherapy (RT) followed by chemotherapy. Post-operative chemotherapy in non-metastatic medulloblastoma was shown to lead to inferior outcomes in the HIT’91 trial [18] and historically, trials incorporating post-operative radiotherapy [5,42] had better outcomes than those using post-operative chemotherapy [21,43].

RT in SIOP PNET5 MB should start within 28 days after first tumour surgery and a delay in the initiation of RT over 40 days post-op renders a patient ineligible for participation in SIOP PNET5 MB. This was based on the observation in PNET4 that a delay in RT resulted in poorer PFS [10]. While in PNET4 the observed cut-off for risk of a lower PFS was 49 days, the cut-off in SIOP PNET5 MB was set to 40 days to include a safety margin.

Radiotherapy in SIOP PNET5 MB is delivered to the entire central nervous system as craniospinal irradiation (CSI), followed by a boost to the tumour region, in once-daily, five weekly fractions of 1.8Gy. The CSI-dose depends on the risk stratum: 18Gy in the LR-Stratum, 23.4Gy in the SR and WNT-HR stratum. RT dose prescription for the SHH-TP53 stratum depends on clinical and biological risk strata (metastasis, TP53-germline alteration).

While in PNET4 the boost to the primary tumour was delivered to the entire posterior fossa, the boost in SIOP PNET5 MB is given to the tumour region only, defined by the contact area between brain structures and tumour plus 1 cm in 3 planes. This has been introduced with the purpose of dose reduction to the temporal lobes, hippocampi, pituitary gland and cochlea [44], and was made possible by increasing access to modern RT techniques, rendering more precise delineations of the target volume using pre and postoperative MRI fusion, and techniques of image guided RT. When SIOP PNET5 MB started, this reduction of the boost volume was solely based on single centre and non-randomised comparisons [45,46]. However, in the meantime, results from the phase III randomised ACNS-0331 trial comparing posterior-fossa irradiation vs. tumour bed boost in clinically standard-risk medulloblastoma have demonstrated the safety of this RT volume reduction [47].

Conformal RT planning and delivery techniques are mandatory in SIOP PNET5 MB, and the use of intensity-modulated radiotherapy (IMRT) or proton beam therapy is encouraged. Currently, approximately 50% of the German patients are receiving proton beam radiotherapy [48].

As reported by the French group, central quality control of RT is paramount and e.g., deviations of doses to the cribriform plate are correlated with a decreased overall survival [49]. Because quality of radiotherapy delivery is considered to be critical, pretreatment quality assurance of RT (RT-QA) is mandatory for SIOP PNET5 MB participants. First experiences of RT-QA have been published from the German and the Italian groups, showing that potentially clinically relevant deviations from the protocol are frequent [48,50,51] and decrease with increasing experience of the treatment site with the protocol [48,50].

### 3.3. Chemotherapy during Radiotherapy

Carboplatin during RT is the randomised question in the SR stratum of PNET5 (see below). No other chemotherapy is given during RT in the LR, SR and WNT-HR strata.

### 3.4. Maintenance Chemotherapy

Chemotherapy plays an important role in childhood medulloblastoma. However, the optimal combination of drugs is unknown, and many different schemes have been in use after radiotherapy with none of them having demonstrated superiority over the other [5,10,18,42,47,52]. In PNET4, eight cycles of ‘Packer’-chemotherapy (VCR 1.5 mg/m^2^, max. dose 2 mg, days 1, 8, 15; lomustine 75 mg/m^2^, day 1; Cisplatin 70 mg/m^2^, day 1; rest until day 42) have been used. However, toxicity of this regimen is substantial and dose reductions are frequently required [18]. Therefore, it was decided to replace every second cycle of chemotherapy by a cyclophosphamide-containing cycle. Cyclophosphamide is an active drug in medulloblastoma and used in various protocols, such as the St. Jude MB trials [4], the HIT-SKK trials [14,41] and other COG-trials [47,53]. Replacing lomustine by cyclophosphamide has been shown to be safe in a randomised trial [5]. Unlike most of the published protocols, in SIOP PNET5 MB the newly introduced ‘B’-block does not contain cisplatin but just vincristine and cyclophosphamide (VCR 1.5 mg/m^2^, max. dose 2 mg, day 1, cyclophosphamide 1.000 mg/m^2^, days 1 and 2, rest until day 22).

The duration of maintenance chemotherapy in SIOP PNET5 MB depends on the stratum, with 8 cycles (4 cycles A and 4 cycles B) in SR, 6 cycles (3 cycles A and 3 cycles B) in LR, six to eight cycles (eight for patients older than 16 and six for patients younger than 16) in WNT-HR (Figure 1, Figure 2 and Figure 3).

During the SIOP PNET5 MB trial, an increased incidence of posterior reversible encephalopathy syndrome (PRES) was observed after the first A cycle, which had not been described in previous studies on medulloblastoma. The cause of the PRES was not identified despite intensive review of all cases. However, PRES is a known side effect of vincristine and all PRES cases occurred in close proximity of the first exposure to vincristine. Since in previous trials, vincristine had always been used during RT we speculated that the first exposure to vincristine in close proximity to CSI might be related to PRES and therefore decided to alter the order of chemotherapy cycles during maintenance chemotherapy. Hereby, the first exposure to vincristine after RT would not be three weekly doses of vincristine, but a single dose followed by a three-week rest.

### 3.5. Chemotherapy in the SHH-TP53 Stratum

The concept of the SHH-TP53 stratum is to assess and intend to improve anti-tumour efficacy of chemotherapy and simultaneously reduce the intensity of therapy, in particular the use of alkylating agents, in order to decrease the risk of secondary malignancies especially in patients with Li-Fraumeni syndrome (LFS) [54]. This reduction of alkylating agents is compensated by increased use of non-genotoxic and p53 function-independent microtubule inhibitors vincristine and vinblastine [55]. Vinblastine was introduced for maintenance chemotherapy because of the decreased risk of peripheral neurotoxicity compared to vincristine.

Another drug expected to have low genotoxicity in TP53 deficient cells while still showing high anti-tumoral effectivity is methotrexate (MTX), so it was decided to use this drug in SHH-TP53 mutant medulloblastoma. Because of the neurotoxic potential of MTX if used after RT, postoperative chemotherapy before RT and use of an alkylator-free, modified HIT-SKK-like chemotherapy, as used in high-risk medulloblastoma in the German MET-HIT2000-AB4 series [56] was introduced in this highly particular subgroup (Figure 3).

## 4. Aims and Design of the Trial Strata

### 4.1. Low Risk Stratum of the SIOP PNET5 MB Study

At the time of the SIOP PNET5 MB study design, it was known that WNT pathway activation defines a unique molecular sub-group of medulloblastomas, which display distinct gene expression profiles, patterns of genomic abnormalities, immunohistochemistry profile and clinical outcome [34,57,58,59,60,61,62,63]. Genomically, WNT-active medulloblastomas appear to be exclusively associated with an isolated loss of an entire copy of chromosome 6 in the majority of cases [25,63,64]. The WNT-active medulloblastoma criteria clearly define a subgroup with good prognosis, and, at the time of SIOP PNET5 MB design, β-catenin status had been shown to be an independent marker of favourable clinical outcome (greater than 90% overall survival) across independent clinical trials-based biological studies [4,27], including the PNET4 study for children younger than 16 years [29].

The methods to assign tumours to the WNT medulloblastoma subgroup in the context of a prospective multicentric international trial have improved over the last 10 years, allowing increased specificity of WNT subgrouping. Early criteria chosen for the SIOP PNET5 MB study were based on immunohistochemical analysis of stabilisation of nuclear β-catenin expression. These have since been refined to incorporate molecular diagnosis either with direct *CTNNB1* (encoding β-catenin) mutation assessment, or assessment of isolated somatic monosomy 6, as recommended in an international consensus [65] and absence of high-risk biological features (*MYC* or *MYCN* amplification). A further amendment during the conduct of the SIOP PNET5 MB study mandated molecular subgrouping by methylome or expression analysis.

### 4.2. Rationale for Lowering the Craniospinal Dose in the LR Stratum of the PNET 5 MB Study

It was demonstrated that WNT-medulloblastoma but no other medulloblastoma subgroups lacks a blood brain barrier [66].

In addition to the favourable clinical outcomes described above, previous studies of the toxicity of craniospinal dose on neurocognitive functions have shown a clear dose–effect relationship. This has been the rationale to decrease the craniospinal dose from 36 to 23.4 Gy in standard risk medulloblastoma, establishing better cognitive outcome after craniospinal dose reduction [9,67]. Furthermore, age < 7 years at diagnosis was the most prominent risk factor for neurocognitive decline and the reason for the Children’s Oncology Group to launch a randomised phase III trial in 2004 to evaluate further dose reduction from 23.4 to 18.0 Gy in children age 3–7 years [9,47]. A recent publication has confirmed a significantly reduced decline in IQ in younger children with 18.0 Gy compared to 23.4 Gy [47]. Although dose reductions to 18.0 Gy are not recommended in this study, in particular due to the high number of leptomeningeal failures primarily driven by group 4 patients, there is no suggestion that exploring the de-intensification strategy in the group of young patients with favourable molecular profile is unwarranted. Consequently, both biological and clinical aforementioned data still justify the decision to set up the single arm SIOP PNET5 MB LR prospective study with craniospinal dose reduction of 18.0 Gy in patient with classical or desmoplastic medulloblastoma, WNT-subgroup, age < 16 years, M0, <1.5 cm^2^ residue and MYC/MYCN negative disease. At present time, at least two prospective trials evaluating doses between 15 and 18 Gy in WNT subgroup patients are ongoing: SJMB12 (NCT01878617) from the St Jude Childrens Research Hospital uses 15 Gy CSI for completely resected, WNT-activated medulloblastoma without further risk factors, and ACNS1422 (NCT02724579) from the Children’s Oncology Group uses 18 Gy CSI for completely resected (residual less than 1.5 cm^2^), WNT-activated medulloblastoma without further risk factors. Omission of radiotherapy as demonstrated in a pilot study exploring a surgery and chemotherapy-only approach, resulted in an unacceptable rate of CNS-relapses and early study closure after enrolling six children [68].

The aim of the SIOP PNET5 MB LR study is to prospectively confirm the high rate of event-free survival in patients between the ages of 3 and 5 years and less than 16 years, with ‘standard risk’ medulloblastoma and a low-risk biological profile after reduced CSI dose, standard radiation dose to the initial tumour bed (54 Gy) followed by reduced chemotherapy using 6 instead of 8 cycles alternating vincristine, cisplatin and CCNU with vincristine and cyclophosphamide (Figure 1). The primary endpoint is the 3-year EFS rate. The aim of the study is to achieve a 3-year rate in excess of 80%. Interim analyses and stopping rules are implemented.

The secondary objectives include investigation of overall survival (OS) rate, progression free survival (PFS), study of late effects (hearing, endocrine, and neurologic function, and standardized, patients/parents rated measurements of health status, executive function, behavioural outcome, and quality of life) and conduct further biological/biomarkers studies to better characterize this subgroup.

### 4.3. Standard-Risk Stratum

According to the PNET 4 study results, the 5-year event-free survival (EFS) for patients without a favourable biological profile, receiving 23.4 Gy in daily fractions of 1.8 Gy on the craniospinal axis, may be around 75% [10]. In this study, the dominant site of relapse is within the craniospinal axis only, suggesting that an intensification of the neuroaxis treatment could improve survival rate. Since salvage strategies at relapse yield poor results, intensification at primary treatment was considered, taking into account the risk of neurocognitive damage and the cumulative toxicity from vincristine and cisplatin leading to dose modifications in the adjuvant setting [18]. Hence the interest arose to explore the role of chemotherapy added during the RT phase of the treatment.

Given the reduced risk of nephron- and ototoxicity, carboplatin was suggested as an attractive platinum analogue. In addition, from a pharmacological point of view, carboplatin more efficiently penetrates into the brain and has a longer free platinum half-life compared to cisplatin [69].

At the time SIOP PNET5 MB was designed, the results of the phase I/II COG 99,701 study were reported, investigating the feasibility of carboplatin given 5 days a week simultaneously to dose-intensive craniospinal and boost irradiation along with weekly vincristine, followed by maintenance chemotherapy in children with newly diagnosed high-risk medulloblastoma [53]. Haematological toxicity was dose-limiting while ototoxicity was acceptable. This regimen resulted in a favourable 5-year PFS rate of 71% and was the basis for the later randomised phase III Children’s Oncology Group ACNS-0332 study [70]. Recently, this study has demonstrated an improvement of EFS by 19% at 5-years for children with high-risk group 3 medulloblastoma receiving concurrent carboplatin during 36 Gy craniospinal irradiation. These results are encouraging for the radio-sensitizing effect and the survival impact of concurrent carboplatin (Figure 2).

Moreover, a significantly lower cumulative dose of cisplatin will be given in the PNET 5 MB-SR stratum during maintenance chemotherapy. In addition, guidelines with stopping rules for the monitoring of the hearing function and neurotoxicity have been implemented in the statistical design of the SIOP PNET5 MB-SR stratum.

Taking into account all aforementioned considerations, a randomised study of the addition of carboplatin during RT in patients eligible for the SR-stratum was deemed acceptable.

The primary objective of this study is to assess whether concurrent administration of carboplatin during radiotherapy results in a better EFS. Sample size calculation is based on assumption of 3-year EFS of 75% for patients with standard therapy. 3-year EFS for patients with additional carboplatin during radiotherapy is assumed to be 10% higher, i.e., 85%. Interim analyses and stopping rules are implemented. The secondary objectives are identical to the ones in the LR-stratum.

### 4.4. WNT-HR Stratum

The WNT-HR stratum (Figure 3) was introduced into the SIOP PNET5 MB trial with the 2nd substantial amendment in 2017. Its introduction was based on the observation that the very rare patients with WNT-activated medulloblastoma and clinical high-risk features had a favourable prognosis when treated with high-risk therapy [4,56]. Therefore, the SIOP PNET5 MB study group considered a reduction of therapy intensity (and therefore long-term side effects) clinically justified. However, a formal confirmatory clinical trial was not possible because WNT-activated medulloblastoma with high-risk features is a very rare disease and only one to two patients per year were expected to be eligible for this trial throughout Europe. Therefore, the WNT-HR stratum was designed as an exploratory stratum to estimate survival rates in these very rare patient group with the aim to explore possibilities for further clinical research.

Children with ‘high-risk’ WNT medulloblastoma younger than 16 years with postoperative residual disease, metastases or high-risk histology are treated postoperatively with 23, 4 Gy CSI, a total of 54 Gy to the tumour bed and boost to metastatic sites, followed by 8 cycles of maintenance chemotherapy as in the SR-stratum.

### 4.5. SHH-TP53 Stratum

The SHH-TP53 stratum (Figure 3) was also introduced with the 2nd substantial amendment in 2017. The background was the observation that SHH-activated medulloblastoma with TP53 mutation carry an extremely high risk of relapse, both if observed in the context of an LFS (germline TP53 mutation) as well as in somatic TP53-mutated SHH-MBs [37]. Unpublished data suggested that treatment intensity did not improve survival in LFS-associated medulloblastoma (T Milde, unpublished), and at the same time patients with LFS have a high risk of second malignancies [71], and the use of alkylating agents is discouraged based on their genotoxicity. Therefore, reduction of therapy and avoidance of alkylators (as compared to standard high-risk therapy protocols) seemed justified.

As for the WNT-HR stratum, the expected number of cases are very small. Therefore, a formal statistical design could not be established for the SHH-TP53 stratum and all analysis were considered exploratory. A secondary aim was to provide a common treatment protocol to allow analysis of a uniformly treated cohort, as analysis of prior case series is severely limited by the use of a wide variety of therapy regimens.

## 5. Biological Investigations: Reference Assessments and Biological Studies

### 5.1. Strategy

The overall strategy for biological investigations within SIOP PNET5 MB studies is two-fold; (i) to use molecular diagnostics of well-defined biomarkers to enrol and stratify patients into the LR and SR study strata, and (ii) to conduct comprehensive studies on the biological basis of medulloblastoma, with the aim of identification, investigation and validation of biomarkers and drug targets with potential to improve management of the disease.

### 5.2. Establishing Practice and Standards

Practices developed through SIOP PNET5 MB have introduced standardised real-time centralised molecular diagnostics and pathology review for medulloblastoma patients across Europe, for the first time. These are supported by the introduction of contemporary practices for the routine collection of high-quality samples (i.e., fresh-frozen and FFPE tumour material, blood (all mandatory) and CSF (optional), essential for clinical and research investigations [72]. A biology and pathology group within the SIOPE embryonal tumours group works to establish, undertake, coordinate and quality control these processes [73], together with translational biological studies, within SIOPE medulloblastoma clinical trials; the committee has representatives from all partner countries.

### 5.3. Diagnostic Criteria

Centralised molecular diagnostics and pathology review must be completed ideally within 3 weeks post-surgery to enable timely planning and commencement of adjuvant therapies. Definition of diagnostic criteria for molecular tests, and quality control/validation of diagnostic methods, are an essential component of the biology group’s work, and have evolved to introduce emerging technologies and methods through protocol amendments. Critical advances have included a requirement for definition of molecular subgroup status by consensus across at least two independent assays (e.g., immunohistochemistry (IHC), direct beta-catenin mutation analysis, DNA methylation or expression profiling), the definition of thresholds for positivity of ‘gold-standard’ iFISH-based testing for *MYC* and *MYCN* amplification status, and the introduction of pathologist panels to review interpretation of IHC analysis [33,73].

### 5.4. Biological Research Questions

Following upfront diagnostic assessments, samples are shipped to designated international research coordinating centres (Newcastle University, UK (for all of Europe); Bonn (for Germany, Austria and Switzerland)). Here, frozen and FFPE tissues are processed, and tissue microarrays (TMAs) constructed to support biological studies. Comprehensive biological studies are performed on surplus collected material by a network of partner research centres, to advance biological understanding of the disease, identify and validate next prognostic and predictive biomarkers. A comprehensive core set of prospective biological investigations is undertaken (RNA-seq, Illumina-850K-copy number/DNA methylation, panel sequencing (tumour/germline) of all commonly-mutated MB genes), alongside establishment of a tissue, TMA and DNA/RNA resource for future planned studies, such as WGS, proteomic and ctDNA (CSF) evaluations.

Integrated biological and genetic datasets obtained are being used, alongside clinical phenotyping, to address key questions and inform planning of future studies, including:

Identification and/or validation of independent prognostic biomarkers which are associated with disease course in LR (i.e., WNT) and SR medulloblastoma.Development of models for the optimal prediction of disease risk, using combined clinical, pathological and molecular indices, within the LR and SR strata.Prioritisation of potential therapeutic targets, and associated predictive biomarkers, for further investigation and validation.Investigation of novel germline predisposition within the cohort.Investigation of associations with clinical factors such as imaging features, quality of survival, intellectual outcomes and toxicity measures.

## 6. Conclusions and Outlook

The eligibility criteria and risk-stratification schemes for SIOP PNET5 MB are based on contemporary understanding of the biological features of medulloblastoma and their clinical relevance. Hence, SIOP PNET5 MB is the first European trial using clinical, histological and molecular parameters for inclusion and stratification of medulloblastoma patients. The newly implemented upfront assessment of molecular parameters requires the availability and timely submission of adequate tumour tissue as an inclusion criterion. This might also allow the investigation and/or validation of further biological parameters which have been discovered during the recruitment of SIOP PNET5 MB and other studies, e.g., medulloblastoma subtypes, including refinement of stratification schemes for risk-adapted treatment.

Complementary to SIOP PNET5 MB, the European trial SIOP HR-MB for children older than 3–5 years with high-risk medulloblastoma is currently implemented in Europe. The diagnostic criteria used in SIOPE HR-MB are equivalent to those used in SIOP PNET5 MB, and are assessed using common pathways, so that most patients with MB within the respective age-group can be allocated to one of the two trials. Planning of the SIOP PNET5 MB successor trial is under way, and will incorporate findings from SIOP PNET5 MB.

## Figures and Tables

**Figure 1 cancers-13-06077-f001:**
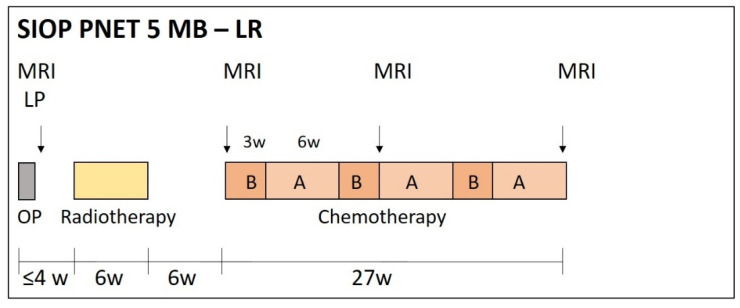
Therapy overview SIOP PNET5-LR stratum. After surgery (OP), patients receive 18.0 Gy craniospinal radiotherapy followed by a boost to 54.0 Gy to the tumour bed. Subsequently, patients receive six cycles of chemotherapy, alternating B (cyclophosphamide 1000 mg/m^2^/d i.v. days 1 and 2 with mesna, vincristine 1.5 mg/m^2^/d day 1, continue with the next block day 22) with A (cisplatin 70 mg/m^2^/d day 1, lomustine 75 mg/m^2^/d day 1 and vincristine 1.5 mg/m^2^/d days 1, 8 and 15, continue with next block day 43).

**Figure 2 cancers-13-06077-f002:**
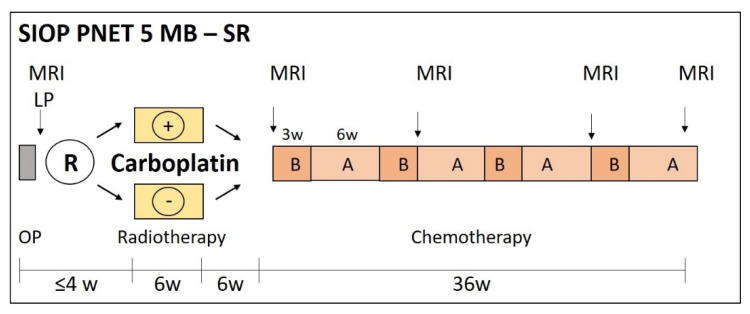
Therapy overview SIOP PNET5-SR stratum. After surgery (OP), patients receive 23.4 Gy craniospinal radiotherapy followed by a boost to 54.0 Gy to the tumour bed. Patients are randomized to receive carboplatin 35 mg/m^2^/d on everyday radiotherapy (i.e., usually 5 days a week). Afterwards, patients receive eight cycles of chemotherapy, alternating B with A (see legend of Figure 1 for drug doses in A and B).

**Figure 3 cancers-13-06077-f003:**
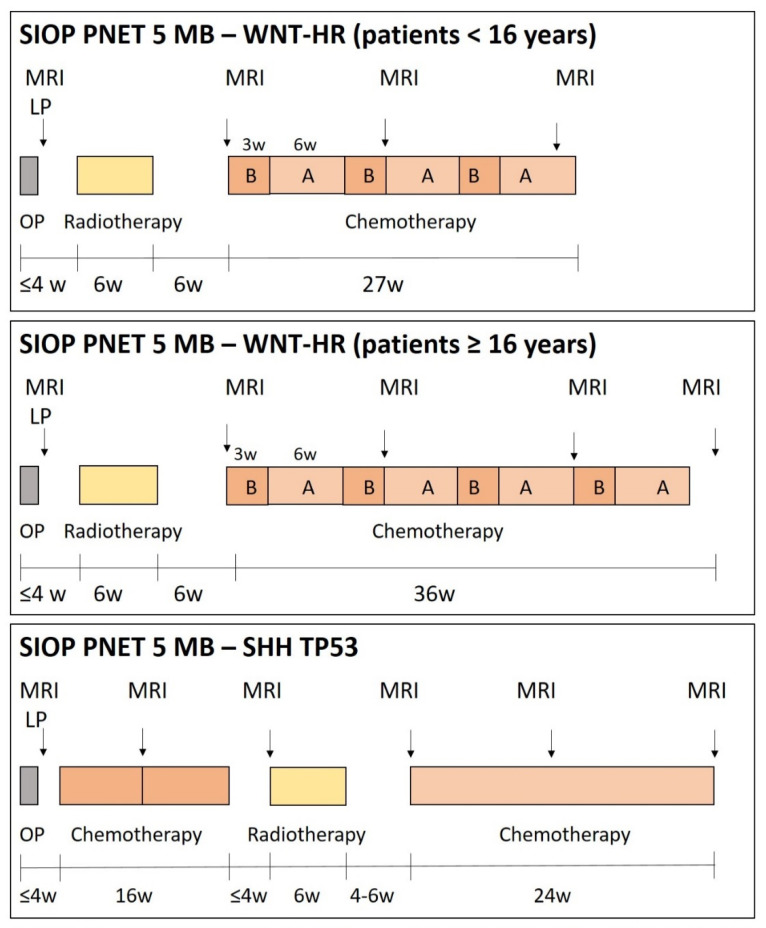
Therapy overview SIOP PNET5-WNT-HR and SHH-TP53 strata. After surgery (OP), patients in the WNT-HR stratum receive 23.4 Gy craniospinal radiotherapy followed by a boost to 54.0 Gy to the tumour bed and boosts to metastatic sites if applicable. Afterwards, younger than 16 receive six, patients 16 years or older at surgery receive eight cycles of chemotherapy, alternating B with A (see legend of Figure 1 for drug doses in A and B). Patients in the SHH-TP53 stratum receive 2 cycles of HIT-SKK like induction chemotherapy with Vincristine 1.5 mg/m^2^/d (max. 2.0 mg), days 1, 15, 29 and 43; Doxorubicin 37.5 mg/m^2^/d, days 1–2, intraventricular Methotrexate 2 mg (via Rickham/Ommaya), days 1–4, 15, 16, 29, 30, 43–46; HD-Methotrexate 5 g/m/d (24 h-infusion, 10% of dose within first 30 min, 90 over 23.5 hours), days 15 and 29; Carboplatin 200 mg/m^2^/d, days 43, 44, and 45. Radiotherapy is stratified according to (a) presence or absence of metastatic disease and (b) presence or absence of germline TP53-alteration (Li-Fraumeni Syndrome). Patients with non-metastatic medulloblastoma and germline TP53-alteration receive focal radiotherapy to the tumour bed to a dose of 54 Gy. Patients with metastatic medulloblastoma and germline TP53-alteration receive 23.4 Gy craniospinal radiotherapy (CSI) with boost to primary tumour bed up to 54 Gy and further boosts to intracranial (54 Gy) and spinal (45 Gy) metastasis. Patients without germline alteration in TP53 receive 36 Gy CSI with a boost to the primary tumour bed up to 54 Gy and further boosts to intracranial (54 Gy) and spinal (45 Gy) metastasis if applicable. Patients receive weekly vincristine (1.5 mg/m^2^/week, max 2.0 mg) up to a maximum of six doses if tolerated. After radiotherapy, patients receive weekly vinblastine (5 mg/m^2^/week, max 10 mg) for 24 weeks with modification to tolerance.

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
