# Peer review of "SIOP PNET5 MB Trial: History and Concept of a Molecularly Stratified Clinical Trial of Risk-Adapted Therapies for Standard-Risk Medulloblastoma"

_cancers, 2021, doi:10.3390/cancers13236077_

Round 1

Reviewer 1 Report

The present manuscript describing the study design and the use of molecular parameters together with clinical and histological methods to stratify therapy for Medulloblastoma patients in SIOP PNET5 MB is relevant and pertinent. The text is clear and appropriate for publication, with minor spelling revision. I believe the name "Stefa Rutkowski" is misspelled in the author's list.

Author Response

Thank you.

The misspelling has been corrected accordingly. 

Reviewer 2 Report

In this article, the authors address the history and concept of SIOP PNET5 MB trial, a molecularly stratified clinical trial of risk adapted therapies for standard risk medulloblastoma. It is quite reasonable that, to improve the prognosis (survival and function) of the patients with medulloblastoma, biological parameters and risk-adapted treatment should be considered in the next step.

This paper may serve as a reference to the reader.

I have some comments.

  • Regarding “Title”, what does “Stage 1” mean?
  • Regarding the terms, it may be slight confusing for readers to use “low-risk (LR)”, standard-risk (SR)” and “high-risk (HR)” strata in “standard-risk” medulloblastoma.

Author Response

Thank you.

The term "Stage 1" in the title was required by the journal and was supposed to be deleted later.

It has already been deleted now for clarity. 

WE agree that the terms LR, SR and HR might be somewhat confusing. However, they reflect the risk attribution for the respective groups, and are also used in the PNET5 trial protocol. Therefore, and as we do not see how to simplify this without losing information, we prefer to leave the terms as written. 

Reviewer 3 Report

Thank you very much for giving me the opportunity to review the work entitled: "SIOP PNET5 MB trial: history and concept of a molecularly-stratified clinical trial of risk-adapted therapies for standard-risk medulloblastoma". The work is very clear and the reading is very fluent, describing the background that led to the conception of the PNET5 trial and the evolution of biological knowledge on medulloblastoma that led to the production of amendments. It is an excellent example of an international collaboration and opportunities for patients who are treated throughout Europe with this protocol. In my opinion, the manuscript is open for publication after some minor revisions that include the following: can the authors justify why the St. Jude approach was not used, which provides for an even greater treatment stratification with respect to the molecular subgroup? The article needs additional editing and in particular should check the spaces between words and the abbreviations; the bibliography should be uniform and two items appear to be overlapping (29 and 32); in paragraph 5.4, the first bullet point has a different character than the rest of the text. Congratulations to the authors on this work.

Author Response

The SJ strategy has not been applied, because the European group preferred to ask a broader randomized question in the PNET5 SR trial (e.g. +- carboplatin), which would not have been possible with more but smaller subgroups (if ranking biological parameters as done in SJ).

Spaces have been checked and corrected.

Bibliography has been done in a uniform way, and lit 32 has been deleted. 

The bullet point in 5.4. has been re-edited.